# Association Mapping for Evaluation of Population Structure, Genetic Diversity, and Physiochemical Traits in Drought-Stressed Maize Germplasm Using SSR Markers

**DOI:** 10.3390/plants12244092

**Published:** 2023-12-07

**Authors:** Muhammad Zahaib Ilyas, Hyeon Park, So Jung Jang, Jungeun Cho, Kyu Jin Sa, Ju Kyong Lee

**Affiliations:** 1Department of Applied Plant Sciences, College of Agriculture and Life Sciences, Kangwon National University, Chuncheon 24341, Republic of Korea; zahaib1996@kangwon.ac.kr (M.Z.I.); hyeonpark@kangwon.ac.kr (H.P.); ttojung24@kangwon.ac.kr (S.J.J.); jjejje2000@kangwon.ac.kr (J.C.); 2Interdisciplinary Program in Smart Agriculture, Kangwon National University, Chuncheon 24341, Republic of Korea; 3Department of Crop Science, College of Ecology & Environmental Sciences, Kyungpook National University, Sangju 37224, Republic of Korea; sakyujin@knu.ac.kr

**Keywords:** microsatellite markers, maize-inbred lines, population structure, physiochemical traits, association mapping, marker–trait association

## Abstract

Globally, maize is one of the most consumed crops along with rice and wheat. However, maize is sensitive to different abiotic stress factors, such as drought, which have a significant impact on its production. The aims of this study were to investigate (1) genetic variation among 41 maize-inbred lines and the relationships among them and (2) significant marker–trait associations (SMTAs) between 7 selected physiochemical traits and 200 simple sequence repeat (SSR) markers to examine the genetics of these traits. A total of 1023 alleles were identified among the 41 maize-inbred lines using the 200 SSR loci, with a mean of 5.1 alleles per locus. The average major allele frequency, gene diversity, and polymorphism information content were 0.498, 0.627, and 0.579, respectively. The population structure analysis based on the 200 SSR loci divided the maize germplasm into two primary groups with an admixed group. Moreover, this study identified, respectively, 85 SMTAs and 31 SMTAs using a general linear model (Q GLM) and a mixed linear model (Q  + K MLM) with statistically significant (*p* < 0.05 and <0.01) associations with the seven physiochemical traits (caffeic acid content, chlorogenic acid content, gallic acid content, ferulic acid content, 2,2-diphenyl-1-picrylhydrazyl free radical scavenging activity, leaf relative moisture content, total phenolic content). These SSR markers were highly correlated with one or more of the seven physiochemical traits. This study provides insights into the genetics of the 41 maize-inbred lines and their seven physiochemical traits and will be of assistance to breeders in the marker-assisted selection of maize for breeding programs.

## 1. Introduction

Maize (*Zea mays* L.) is an extensively grown multi-purpose cereal crop with a dynamic and diverse role in the overall agri-food system [1]. It serves as food for humans, fodder for livestock, and a key raw material for processed corn-based products such as corn starch and corn syrup [2]. Moreover, maize is used as biomass for bioenergy production and serves as a model plant for genetics and genomic studies because of its flexible and diverse genetics [2]. It also serves as a source of phytochemicals such as phenolics, and their concentration differs among genotypes [3].

Plants in agricultural ecosystems often encounter various abiotic stresses such as drought, salinity, and heavy metals [4]. Among these environmental constraints, drought stress has a substantial effect on maize growth by interrupting plant morphological, biochemical, and physiological processes [5]. Excessive water loss from leaves during drought conditions disrupts the electron transport chain, leading to increased generation of reactive oxygen species (ROS) in plants [6]. The elevated level of ROS damages cellular structures by degrading cell proteins, lipids, and nucleic acids, eventually resulting in cell death [7].

Conversely, plants possess a highly developed antioxidant defense system to cope with oxidative stress resulting from excessive ROS production [8]. This defense system consists of both enzymatic, i.e., peroxidase (POD), superoxide dismutase (SOD), ascorbate peroxidase (APX), etc., and non-enzymatic, i.e., phenolics, ascorbic acid (AsA), etc., antioxidants, which work in tandem to minimize the lethal effects of oxidative stress [8]. Among non-enzymatic antioxidants, phenolic compounds play a vital defense role under unfavorable environmental conditions [9].

Phenolic compounds are secondary metabolites present in plant tissues [10], and their excessive accumulation is recognized as a unique trait associated with plant stress response [9]. A phenolic compound functions as a nucleophile, engaging with free oxygen radicals such as hydroxyl ion, superoxide, and lipid peroxyl radicals [11], which hinders lipid peroxidation by scavenging free radicals and averting damage [12]. Robards and Antolovich [13] stated that about 2% of all photosynthesized carbon is converted into plant phenolics. The phenolic compounds found in cereals are mostly phenolic acids that are derivatives of cinnamic acids (caffeic acid, ferulic acid, etc.) and benzoic acids (gallic acid, salicylic acid, etc.) [14,15]. The antioxidant potential of phenolic acids found in cereals declines in the following order: gallic, caffeic, benzoic, sinapic, syringic, ferulic, p-coumaric, vanillic, and 4-hydroxybenzoic acid [14,15].

With the increasing frequency and intensity of drought occurrences resulting from climate change, genetic improvement of maize is imperative for augmenting its drought tolerance [16]. Drought tolerance results from the quantitative inheritance of complex morphological, biochemical, and physiological traits [17]. In conventional breeding, phenotypic characterization is indispensable for advancing breeding programs and is labor-intensive, time-consuming, and greatly influenced by environmental factors [18]. Conversely, molecular characterization is unaffected by environmental factors and provides useful and accurate complementary genetic information [19]. There has been extensive application of molecular markers, including mapping of quantitative trait loci (QTL), gene cloning, studying of marker–trait associations, genetic diversity, population structure, and marker-assisted selection (MAS) for genetic and breeding research [20,21]. MAS relies on molecular markers to select superior phenotypic traits based on genotypes, and for its effectiveness, the significance of identifying linked markers and their corresponding genetic regions has been highlighted [22]. MAS, therefore, requires the mapping of high-density QTL in populations with desired traits, which is time-consuming and demands considerable effort, prompting the search for alternative approaches to reduce the time and effort required [23]. Recently, marker–trait association mapping, utilizing linkage disequilibrium, has enabled breeders to determine precise associations between markers and complex phenotypic traits and has three additional advantages: enhances mapping resolution, decreases research time and effort, and enables investigation of large numbers of alleles compared with QTL mapping [23,24].

Among all DNA-based molecular markers, simple sequence repeat (SSR) markers, also called microsatellites, offer valuable insights into genetic relationships, population structure, and genetic diversity in a population because of their high polymorphism, reproducibility, codominance, and extensive distribution within the plant genome [25,26]. These microsatellites have been used successfully for genetic diversity, population structure, and marker–trait association studies in maize [27,28,29,30]. Favorable alleles that contribute to abiotic stress tolerance, higher yield, and nutritional quality improvement are present in maize germplasm; however, they are typically dispersed across a variety of landraces or populations [2].

The genetic diversity of maize and marker–trait associations have received considerable attention, primarily aimed at enhancing yield and nutritional quality. However, to the best of our knowledge, there has been limited investigation into maize genetics with regard to secondary metabolites, particularly phenolic compounds, and the identification of connections between SSR markers and these phenolic compounds. The purpose of this study was to analyze genetic diversity, population structure, and significant marker–trait associations (SMTAs) between maize physiochemical traits and SSR markers to enable examination of the genetics of these traits. The marker–trait association analysis was performed using 200 SSR markers and 7 physiochemical traits, viz., caffeic acid content, chlorogenic acid content, gallic acid content, ferulic acid content, 2,2-diphenyl-1-picrylhydrazyl (DPPH) free radical scavenging activity, leaf relative moisture content (LRMC), and total phenolic content (TPC). Finally, we searched the maize genome database and previously reported literature to determine whether the identified SSR markers were co-localized with candidate genes/QTLs.

## 2. Results

### 2.1. Physiochemical Traits and Their Correlation Analysis

The results from seven physiochemical traits studied in 41 maize-inbred lines under drought stress are exhibited in Appendix A. These data have been studied previously by Ilyas et al. (2023) [31]. The caffeic acid content ranged from 0.000 to 2.915 µg/mL across the 41 maize-inbred lines, with a mean value of 1.295 µg/mL. Likewise, the chlorogenic, gallic, and ferulic acid contents ranged from 0.000 to 5.127, 2.064 to 5.414, and 0.000 to 1.833 µg/mL, with mean values of 1.96422, 3.64861, and 1.05082 µg/mL, respectively. TPC, DPPH free radical scavenging potential, and LRMC varied from 55.112 to 179.199 GAE/g sample, 0.000 to 22.759%, and 53.089 to 77.766%, with mean values of 103.999 GAE/g sample, 4.037%, and 69.134%, respectively.

Moreover, a correlation analysis was also conducted to determine the relationships among the seven physiochemical traits (Table 1). A highly statistically significant (*p* < 0.01) positive correlation was observed between caffeic acid and TPC, ferulic acid and gallic acid, and ferulic acid and TPC. Furthermore, at *p* < 0.05, a statistically significant positive correlation was also noted between chlorogenic acid and TPC. Conversely, a negative correlation was recorded between chlorogenic acid and DPPH, chlorogenic acid and gallic acid, DPPH and gallic acid, and DPPH and LRMC.

### 2.2. Genetic Diversity Index

A total of 200 SSR markers were utilized to study genetic diversity among the 41 maize-inbred lines including gene diversity (GD), major allele frequency (MAF), and polymorphism information content (PIC). A total of 1023 alleles were recognized among the 41 maize-inbred lines using the 200 SSR markers (Appendix A). The 200 SSR markers employed in this study were found to be distributed across the ten chromosomes of maize, with a minimum of 17 markers on chromosome 2 and a maximum of 30 markers on chromosome 7. The allele counts per locus ranged from 2 to 11, with a mean of 5.1 alleles per locus (Table 2 and Appendix A). Moreover, the average MAF was found to be 0.498, with a minimum of 0.220 and a maximum of 0.951. The GD values ranged from 0.094 to 0.857, with an average value of 0.627. Furthermore, the PIC ranged between 0.092 and 0.841, with a mean PIC of 0.579 (Table 2 and Appendix A).

### 2.3. Population Structure and Clustering of Korean Maize-Inbred Lines

This study employed ad hoc measures for ∆K, as suggested by Evanno et al. (2005) [32], to tackle the difficulties related to the interpretation of actual K values. The maximum ∆K value for the 41 maize-inbred lines utilizing the 200 SSR markers was found at *K* = 2, and all inbred lines were distinctly classified into two groups **(Figure 1**). Group I had the most inbred lines (29), followed by Group II and the Admixed Group with six maize-inbred lines each. The inbred lines in Group II were 17CS8006, HCW5, HCW4, 16CLP40, HW3, and HW19, whereas the inbred lines in the Admixed Group were 16S8068-9, 15RS8056, 17CS5047, 15S8021-3, 14S8025, and HCW2. The remainder of the 41 inbred lines were categorized within Group I of the population structure (Figure 1).

The genetic relationships among the 41 maize-inbred lines were studied using an unweighted pair group method with arithmetic mean (UPGMA) analysis to categorize the inbred lines into appropriate groups. Through the UPGMA analysis, a distance-based dendrogram was constructed by employing the 200 SSR loci (Figure 2). All 41 maize-inbred lines were categorized into two main groups with a genetic similarity of 33%. Group I was subdivided into two groups, namely Group I-1 and Group I-2. Group I-1 comprised 14 inbred lines (11BS8016-7, 15RS8002, 12S8052, 17YS6032, HW8, KL103, HW4, HW1, HW7, 12BS5076-8, 15RS8039, HW17, HW18, and HW11), among which 2 inbred lines (HW17 and HW18) had the highest genetic similarity of 90%. The inbred lines of Group I-1 exhibited drought-sensitive (11 inbred lines) and moderately drought-tolerant (3 inbred lines) characteristics. Group I-2 consisted of 20 inbred lines (14S8025, 16CLP40, HCW4, HW3, 17CS8006, HCW5, HW19, HCW2, 17CS5047, 15S8021-3, 16S8068-9, 15RS8056, 16CLP23, HCW1, 17CS8067, HCW3, HW9, HW12, KW7, and HW10), and these inbred lines showed drought-sensitive (10 inbred lines), moderately drought-tolerant (7 inbred lines), and drought-tolerant (3 inbred lines) characteristics. Group II comprised seven inbred lines (17YS8003, HW16, HW15, GP3, GP5, HF12, and HF22), which showed drought-sensitive (5 inbred lines) and moderately drought-tolerant (2 inbred lines) characteristics. In summary, drought stress tolerance analysis of the 41 maize-inbred lines showed that 26 inbred lines exhibited drought sensitivity, 12 inbred lines exhibited moderate drought tolerance, and the remaining 3 inbred lines exhibited drought tolerance (Figure 2).

### 2.4. Association Analyses Using Q GLM and Q + K MLM for Seven Physiochemical Traits

Association mapping analysis was performed between the set of 200 SSR markers and 7 physiochemical traits in the 41 maize-inbred lines using a population structure (Q) general linear model (GLM) and a Q + K (kinship) mixed linear model (MLM) to identify statistically significant associations between markers and traits at *p* < 0.05 and <0.01. This study detected a total of 85 SSR markers significantly (*p* < 0.05 and <0.01) associated with the seven physiochemical traits by applying the Q GLM analysis (Table 3). Among these 85 SSR markers, 11 markers were associated with caffeic acid, 14 markers were associated with chlorogenic acid, 17 markers were associated with DPPH, 9 markers were associated with ferulic acid, 29 markers were associated with gallic acid, 16 markers were associated with LRMC, and 12 markers were associated with TPC. In addition, among these 85 SSR markers, 13 markers (umc1030, umc1863, umc2160, umc2185, umc2356, umc1034, umc2026, umc2396, bnlg619, umc1134, umc1805, umc1949, and bnlg1759) were individually associated with two different physiochemical traits. In detail, markers umc1030 and umc2356 were each associated with both caffeic acid and ferulic acid, umc1863 and umc2185 were each associated with both gallic acid and caffeic acid, umc2160 was associated with DPPH and caffeic acid, and umc1034 was associated with chlorogenic acid and TPC. Further, umc2026 was associated with chlorogenic acid and ferulic acid, umc2396 was associated with gallic acid and chlorogenic acid, bnlg619 was associated with DPPH and gallic acid, umc1134 was associated with DPPH and LRMC, umc1805 and umc1949 were each associated with both gallic acid and ferulic acid, and bnlg1759 was associated with TPC and ferulic acid. Furthermore, among these 85 SSR markers, 5 SSR markers (umc2159, bnlg1350, phi101049, umc1576, and umc1906) were each associated with three different physiochemical traits. That is, umc2159 was associated with caffeic acid, DPPH, and TPC; bnlg1350 was associated with chlorogenic acid, LRMC, and TPC; phi101049 was associated with chlorogenic acid, LRMC, and gallic acid; umc1576 was associated with DPPH, gallic acid, and LRMC; and umc1906 was associated with gallic acid, ferulic acid, and TPC. In our study, phenotypic variation was assessed by measuring the coefficient of determination (R^2^) value in a GLM analysis for each SSR marker among the 85 significant marker–trait associations (SMTAs) (Table 3). The highest R^2^ value (53.2%) among the 85 SMTAs was observed for umc2160 linked with DPPH, and the lowest R^2^ value (10.2%) was observed for umc1363 linked with caffeic acid.

Meanwhile, in the Q + K MLM analysis, a total of 31 SSR markers had statistically significant associations with the seven physiochemical traits (*p* < 0.05 and <0.01) (Table 4). Among these 31 SSR markers detected in the Q + K MLM analysis, 1 marker (umc1363) was associated with caffeic acid, 2 markers (bnlg1350 and bnlg1484) were associated with chlorogenic acid, 5 markers (bnlg1079, bnlg1526, umc1063, umc2160, and umc2301) were associated with DPPH, 5 markers (bnlg1246, umc1906, umc1949, umc2026, and umc2356) were associated with ferulic acid, 2 markers (umc1139 and umc2246) were associated with gallic acid, 10 markers (phi101049, umc1134, umc1239, umc1608, umc1627, umc1667, umc1671, umc1934, umc1959, and umc2343) were associated with LRMC, and 7 markers (bnlg1887, phi193225, umc1034, umc1330, umc1388, umc1519, and umc1906) were associated with TPC (Table 4). Among these 31 SMTAs, only 1 marker was associated with more than one trait, with umc1906 being related to both the ferulic acid and TPC traits. In addition, the highest R^2^ value (48.1%) was observed for umc2160 associated with DPPH, while the lowest R^2^ value (10.4%) was for umc1363 associated with caffeic acid (Table 4).

## 3. Discussion

In this study, we focused on seven physiochemical traits of maize (caffeic acid, chlorogenic acid, DPPH, ferulic acid, gallic acid, LRMC, and TPC) and investigated SMTAs using 200 SSR markers. Phenolic compounds contribute in various ways to drought tolerance in plants by liberating hydroxyl groups (OH), oxidizing themselves, and acting as antioxidants, and their antioxidative potential is determined by the number of available hydroxyl groups. For instance, caffeic acid contains two hydroxyl groups, exhibits robust antioxidant properties, and contributes to enhanced drought tolerance [43]. The exogenous application of caffeic acid to *Cucumis sativus* seedlings under drought stress dramatically reduced levels of hydrogen peroxide, superoxide radicals, and malondialdehyde in comparison with the control (no caffeic acid under drought stress). Moreover, caffeic acid increased the activity of catalase (CAT), superoxide dismutase (SOD), ascorbate peroxidase (APX), etc., in contrast with the control [44]. Similarly, when maize seedlings were treated with ferulic acid, the activity of CAT, peroxidase, and indole-3-acetic acid (IAA) substantially increased while the activity of polyphenol oxidase was decreased [45]. The application of gallic acid to drought-stressed rice seedlings conferred protection against oxidative stress by reducing ROS levels, stimulating enzymatic antioxidant activities, optimizing plant water relations, and improving photosynthetic efficiency, thereby strengthening the resilience of plants to drought stress [46].

This study analyzed the correlation between these seven traits and found both positive and negative correlations among the traits (Table 1). Laddomada et al. (2021) [47] found a statistically significant (*p* < 0.05) positive correlation between ferulic acid and total phenolics in durum wheat cultivars, which supports the correlation results in the current study regarding TPC and ferulic acid. Likewise, Hodaei et al. (2018) [48] reported a positive correlation between DPPH free radical scavenging activity and TPC in drought-stressed *Chrysanthemum morifolium*, corroborating the findings of the correlation between TPC and DPPH in the current study. Another study described a strong positive correlation between TPC and caffeic acid in water-stressed *Rosa damascena* [49], aligning with the correlation observed between TPC and caffeic acid in our study (Table 1). In drought-stressed wheat, the LRMC and TPC were positively correlated [50], consistent with the correlation of LRMC and TPC that we observed in our study. Also, there were statistically significant (*p* < 0.05 and 0.01) positive correlations between caffeic acid and ferulic acid, as well as between gallic acid and chlorogenic acid, in salt-stressed wheat, with a negative correlation between caffeic acid and gallic acid [51]. A negative correlation between DPPH and RLMC was reported in drought-stressed *Fagopyrum tataricum* [52] and *Hibiscus sabdariffa* [53], which supports the correlation observed between DPPH and RLMC in our study. In general, the antioxidant activity assessed through the DPPH assay and the levels of different phenolic acids in drought-stressed maize exhibited a positive correlation with TPC, suggesting that higher TPC and phenolic acid levels are associated with increased ROS scavenging in drought-stressed maize (Table 1).

In addition to the physiochemical trait evaluation of the 41 maize-inbred lines, we employed two statistical methods: a UPGMA dendrogram with NTSYS-pc V2.1 and a model-based process with STRUCTURE 2.3 software, to assess precisely the information on genetic diversity and population structure. In this study, the analysis of the population structure pattern using STRUCTURE 2.3 software with the 200 SSR markers revealed that the maize population comprised two main groups and an admixed group. Group I had the highest number of inbred lines (29), followed by Group II and the Admixed Group, which each consisted of six maize-inbred lines (Figure 1). Our results are consistent with a study conducted by Mathiang et al. (2023) [29], which examined the population structure of 37 South Sudan maize landrace accessions using STRUCTURE 2.2 software. They also identified two main groups, Group I and Group II, and one admixed group. Group I comprised 20 accessions representing 54.1% of the total maize population. Group II comprised 12 accessions representing 32.4% of the population, and the admixed group included 5 accessions [29]. Likewise, Sa et al. (2023) [30] investigated the population structure of 80 maize-inbred lines using STRUCTURE 2.3 software. They revealed that all the inbred lines were divided into two primary groups (Group I and Group II) and one admixed group. Group I contained 28 lines, Group II comprised 16 lines, and the admixed group included 36 lines [30]. Moreover, a study conducted by Kim et al. (2021) [54] on 12 Korean waxy maize-inbred lines using STRUCTURE 2.2 software revealed that the entire maize population under study was categorized into two main groups (Group I and Group II). Group I had seven inbred lines, while Group II included five inbred lines [54]. Furthermore, a phylogenetic tree was constructed employing the UPGMA analysis, based on the 200 SSR markers, to investigate the genetic relationships among the 41 maize-inbred lines (Figure 2). These inbred lines were classified into two major groups at a genetic similarity of 33%. Group I was further subdivided into two groups, namely Group I-1 and Group I-2. Group I-1 comprised 14 inbred lines, with 2 of the inbred lines (HW17 and HW18) displaying the highest genetic similarity of 90%. Additionally, Group I-2 consisted of 20 inbred lines, and Group II consisted of 7 inbred lines. According to a previous study conducted on 22 maize-inbred lines using a UPGMA dendrogram based on 17 SSR markers, the whole maize population was divided into three main clusters at a genetic similarity of 43% [55], which can be related to the results of the current study. In our study, the UPGMA dendrogram clustering pattern could not clearly distinguish the inbred lines into separate groups representing drought-sensitive, moderately drought-tolerant, and drought-tolerant. However, the clustering pattern provided a general overview of their response to drought stress. Briefly, among the 14 inbred lines included in Group I-1, 11 (11BS8016-7, 15RS8002, 17YS6032, HW8, KL103, HW1, HW7, 12BS5076-8, HW17, HW18, and HW11) showed drought-sensitive traits, while only 3 (12S8052, HW4, and 15RS8039) displayed moderately drought-tolerant traits. In Group I-2, out of 20 inbred lines, 10 (HCW4, HW3, 17CS8006, HW19, 17CS8067, HCW3, HW9, HW12, KW7, and HW10) exhibited drought-sensitive traits, 7 (14S8025, HCW5, HCW2, 17CS5047, 16S8068-9, 16CLP23, and HCW1) were moderately drought-tolerant, and only 3 (16CLP40, 15S8021-3, and 15RS8056) demonstrated a substantial level of the seven physiochemical traits, which renders them as potential lines for selection as drought-tolerant inbred lines. Finally, Group II consisted of seven inbred lines, out of which five (17YS8003, HW16, HW15, HF12, and HF22) showed drought-sensitive traits, and two (GP3 and GP5) showed moderately drought-tolerant characteristics. Thus, this clustering pattern depicted that Group I-1 and Group II contained comparatively higher numbers of drought-sensitive lines than Group I-2. Most importantly, 3 of the 41 inbred lines that were screened as drought-tolerant based on their results for the seven physiochemical traits all belonged to Group I-2, which also contained seven other moderately drought-tolerant lines. Hence, it can be summarized that most of the maize-inbred lines that showed relatively better drought-tolerant traits were in Group I-2.

Data pertaining to population structure, genetic diversity, and genetic relationships among the maize-inbred lines can optimize germplasm management and enhance the efficiency of future breeding programs. Genetic diversity refers to the range of genetic characteristics within a given population, and a high degree of genetic diversity is critical for the development of novel cultivars with high adaptability to changing environmental conditions [56]. This study assessed genetic diversity, including GD, MAF, and PIC, among 41 maize-inbred lines using 200 SSR markers. A total of 1023 alleles were identified across 41 maize-inbred lines by employing 200 SSR loci (Appendix A). The number of alleles per locus varied between 2 and 11, with a mean of 5.1 alleles per locus (Table 2 and Appendix A), which is higher than the mean allele count per locus detected in previous maize genetic diversity studies [57,58]. The high mean number of alleles per locus can be ascribed to substantial genetic diversity within the population under investigation. The average MAF was found to be 0.498, with a minimum of 0.220 and a maximum of 0.951. The GD values varied between 0.094 and 0.857, with a mean value of 0.627, which is similar to the GD (0.62) recorded in a subtropical maize population study using SSR markers [59]. Finally, the PIC ranged from 0.092 to 0.841 with an average PIC of 0.579 (Table 2 and Appendix A). The PIC value offers a better assessment of genetic diversity compared with relying solely on the mean allele count per locus as PIC considers the relative frequencies of each allele within the population [60,61]. In comparison with some previous studies, the above-mentioned results of the allele count per locus, GD, and PIC indicated that the Korean 41 maize-inbred lines possess an adequate range of genetic variation. Hence, the high genetic diversity of the 41 maize-inbred line population as determined by the genetic diversity index and structure analyses indicated the stability of the population for association analyses.

After the evaluation of 41 maize-inbred lines for physiochemical traits and genetic characteristics, we examined the association between the selected traits and the markers used to study the genetic characteristics of the maize population. We performed association analyses between 200 SSR markers and seven physiochemical traits using a Q GLM and a Q + K MLM to identify the SMTAs at *p* < 0.05 and <0.01. A total of 85 SSR markers were statistically significantly (*p* < 0.05 and <0.01) associated with the seven traits using the GLM (Table 3). Out of these 85 markers, 13 were individually associated with two distinct physiochemical traits, and 5 SSR markers were each linked to three different physiochemical traits. Moreover, phenotypic variation was determined by calculating the coefficient of determination (R^2^) using the GLM analysis for each SSR marker within the set of 85 SMTAs (Table 3). Using the Q + K MLM analysis, 31 markers were statistically significantly (*p* < 0.05 and <0.01) associated with the seven traits (Table 4). In a study conducted on rice by Sanghamitra et al. (2022) [62], three SMTAs (*p* < 0.01) were detected between DPPH and markers RM247, RM3701, and RM13600 with both GLM and MLM analyses. The highest R^2^ values were attributed to marker RM3701 linked to DPPH, with values of 9% in the GLM and 10% in the MLM analyses [62]. The R^2^ values observed in our study are substantially higher than those reported by Sanghamitra et al. (2022) [62]. In our study, we detected five SMTAs linked to DPPH, with umc2160 exhibiting the highest R^2^ value (53% in the GLM and 48.1% in the MLM analyses) (Table 4). In another study on SSR marker associations with rice physiochemical traits, 4 out of 50 markers were associated with TPC in a GLM analysis, with the RM225 marker exhibiting the highest R^2^ value of 20.8% among these 4 SMTAs [63]. However, we found seven markers associated with TPC with these seven SMTAs detected with both the GLM and MLM. Among these, marker umc1519 exhibited the highest R^2^ value in the GLM analysis of 48.9%, and umc1034 displayed the highest R^2^ value in the MLM analysis of 43.9% (Table 4). Furthermore, Sharma et al. (2020) [64] identified four SSR markers associated with ferulic acid in wheat varieties, with the Fera.gwm296_2A marker showing the highest R^2^ value of 18.7% in a GLM analysis. This study also reported four SSR markers linked to chlorogenic acid, with the Chla.wmc494_6B marker exhibiting the highest R^2^ value of 33% in the GLM analysis [64]. In contrast, we identified two SMTAs for chlorogenic acid and five SMTAs for ferulic acid, which were detected in both the GLM and MLM analyses. For chlorogenic acid, the marker bnlg1350 exhibited the highest R2 values of 49.1% and 46.5% in the GLM and MLM analyses, respectively. The marker umc1949 displayed the highest R2 values of 37.5% in the GLM analysis and 40.4% in the MLM analysis for ferulic acid (Table 4).

Based on a search of the maize genome database (http://www.maizeGDB.org/, accessed on 1 March 2022) or in former linkage-mapping studies, all the SMTAs identified by the GLM and MLM analyses in this study have been linked with genes/QTLs (except bnlg1350 marker). In our study, the marker umc1363 linked to caffeic acid has been associated with the *expansion-like6* gene, as per MaizeGDB (http://www.maizeGDB.org/, accessed on 1 March 2022) (Table 4). The present study identified two SMTAs for chlorogenic acid. The locus bnlg1484 was associated with the *serine/threonine-protein kinase WAG1* gene located on chromosome 1. However, the other locus, bnlg1350, has not been well documented in relation to the mapping of the QTL/gene located at this locus (Table 4). We identified five SMTAs related to DPPH in this study. Three SSR markers, namely, bnlg1079, bnlg1526, and umc2301, were associated respectively with QTLs of vanillin and syringaldehyde [33], the QTL for whole phosphorus utilization of plant under a phosphorus deficit condition [34], and the flanking QTL of the alanine aminotransferase (AlaAT) enzyme [35] in maize. The markers umc1063 and umc2160 were located at a locus containing the *lea protein group3* and *rhicadhesin receptor* genes, respectively. Likewise, a total of five SMTAs were identified in our study for the ferulic acid trait (Table 4). Four of these markers, namely, umc1906, umc1949, umc2026, and umc2356, were located respectively at loci containing the *nucleobase:cation symporter1*, *glutaredoxin family*, *glutathione transferase17*, and *heavy metal transport/detoxification superfamily protein* genes. The marker bnlg1246 was reported as a part of a cluster of QTLs linked to relative above-ground biomass dry weight, relative leaf age, and tolerance index to low phosphorus traits in maize [36]. For the gallic acid trait, we only identified two SMTAs, for markers umc1139 and umc2246, which were reported to be located at loci containing the *small kernel 501* and *phosphomannomutase1* genes, respectively. The maximum number of ten SMTAs identified in this study was for the LRMC trait (Table 4). Four markers, namely, phi101049, umc1134, umc1239, and umc1627, were reported as the *pyridoxin biosynthesis protein ER1*, *Rop guanine nucleotide exchange factor 1*, *Trihelix-transcription factor 43*, and *pathogenesis-related protein17* genes, respectively. Further, five out of the ten markers, namely, umc1667, umc1671, umc1934, umc1959, and umc2343, were found as the *trehalose-6-phosphate phosphatase4*, *protein arginine methyltransferase3*, *OSJNBa0070C17.17-like protein*, *ATP-dependent DNA helicase Q-like SIM*, and *Ran-binding protein 1* genes, respectively. Among the ten markers, umc1608 was identified as a flanking marker by Qu et al. (2020) [37] related to QTLs of total endosperm aborted kernels and total aborted kernels (TA) traits in maize (Table 4). Finally, the present study identified seven SMTAs related to the TPC trait. The marker umc1906 was found to be a *nucleobase:cation symporter1* gene, while the related/near loci for umc1388 were associated with *early response to dehydration 15-homolog3*. The marker bnlg1887 was identified as a QTL associated with ear row numbers [38], and marker phi193225 was reported to be associated with QTL for days to pollen shed [39]. The other three of the seven markers, namely, umc1034, umc1330, and umc1519, were reported to be linked respectively to QTLs of gray leaf spot disease [40], the uronic acids [41], and the root dry weight, shoot dry weight, and total dry weight under waterlogging stress in maize [42].

## 4. Materials and Methods

### 4.1. Seed Collection

In this study, we used a set of 41 maize-inbred lines developed by and obtained from the Maize Experimental Station, Gangwon Agricultural Research and Extension Service, Hongcheon, South Korea. All the maize-inbred lines used in this study were developed through self-pollination. The majority of these maize-inbred lines originated from waxy maize, but there were also some derived from popcorn and flint maize. Numerous F_1_ hybrids of Korean varieties have been developed using these maize-inbred lines as parental lines (Appendix A).

### 4.2. Crop Husbandry and Physiochemical Trait Evaluation

The seven physiochemical traits of the 41 drought-stressed maize-inbred lines used in this study to perform marker–trait association (Appendix A) were previously investigated by Ilyas et al. (2023) [31]. Briefly, a pot study was performed in a glass house under drought stress condition at 30% field capacity (FC) in autumn 2022 (October–November). Plants were grown under normal water conditions (80% FC) until trifoliate stage, and drought conditions (30% FC) were imposed as the collar of the third leaf appeared. The drought stress lasted until the plants were harvested, which was performed on the 21st day of drought stress. The study was carried out in a completely randomized design (CRD) and replicated thrice.

Leaf relative moisture content (LRMC) was estimated by the following equation [65]:$$LRMC\;\left(\%\right)=\frac{LFW-LDW}{LTW-LDW}\times 100,$$
where LFW represents the leaf fresh weight, LDW represents the leaf dry weight, and LTW represents the leaf turgid weight.

To estimate the DPPH free radical scavenging potential, we followed the protocol used by Choi et al. (1993) [66] with minor modifications. We prepared 0.1 g of dried plant powder for testing by adding 4 mL of methanol (MeOH) and diluting it a further 10 times. A 0.1 mL aliquot of a 0.15 mM DPPH solution was mixed with 0.1 mL of diluted plant extract in 96-well plates, followed by incubation in darkness for 30 min. The absorbance at 515 nm was assessed using an ELISA reader (model 680, Bio-Rad Inc., Hercules, CA, USA) to estimate the DPPH radical scavenging activity, employing the following equation [67]:$$DPPH\;free\;radical\;scavenging\;potential\;\left(\%\right)=[1-\left(AbS-AbC\right)]\times 100,$$
where AbS represents the absorbance of the test sample, and AbC represents the absorbance of the control.

The estimation of TPC was conducted with a slight modification of the Folin–Ciocalteu reagent method [68]. A 0.1 mL aliquot of MeOH extracted dried plant sample was added in 0.05 mL of Folin–Ciocalteu reagent, followed by the addition of 0.3 mL of 20% sodium carbonate and 1 mL of distilled water. This mixture was incubated for 20 min at normal room temperature. Subsequently, the absorbance of blank sample was recorded at 725 nm using an ultraviolet (UV)/visible (VIS) spectrophotometer (V530, Jasco Co., Tokyo, Japan). The TPC was determined by constructing a standard calibration curve utilizing gallic acid solutions within the range of 10-250 µg/mL as a standard. The TPC was expressed as milligrams of gallic acid equivalent per gram of the sample (mg GAE/g sample).

Analysis of different phenolic acids was conducted using high-performance liquid chromatography-UV (HPLC-UV) [69]. In brief, the HPLC analysis was carried out through an Agilent 1260 series instrument equipped with a Shiseido (Tokyo, Japan) Capcell Pak C18 column. Numerous phenolic acids were quantified utilizing their respective standard solutions, including chlorogenic acid, gallic acid, ferulic acid, p-coumaric acid, caffeic acid, and syringic acid. The mobile phase comprised a mixture of 0.1% phosphoric acid in water and acetonitrile, with a flow rate set at 1 mL/min. The wavelength of the UV detector was adjusted at 270 nm. Each phenolic acid was identified by matching its retention time with the corresponding standard. Quantification of phenolic acid was accomplished by generating standard curves at concentrations of 0.5, 1, 5, 10, 25, 50, and 100 ppm using external standards. The results were expressed in micrograms per milliliter (µg/mL).

### 4.3. DNA Extraction and SSR Amplification

Genomic DNA for the 41 maize-inbred lines was extracted from the young leaf tissues of each inbred line following the Dellaporta et al. (1983) [70] protocol with slight modifications. A set of 200 SSR markers was employed for analyzing the population structure, genetic diversity, and marker–trait associations for the 41 maize-inbred lines. Information regarding SSR markers, including their location, chromosome number, etc., was acquired from MaizeGDB (http://www.maizegdb.org/, accessed on 17 January 2023). SSR marker amplification was carried out utilizing the EX Taq polymerase chain reaction (PCR) kit (Takara, Ohtsu, Japan). The total volume of reaction mixture was 20 µL, comprising 20 ng genomic DNA, 0.5 µM forward and reverse primers, 0.2 mM dNTPs, 1×EX Taq buffer, and 1 unit of EX Taq polymerase for polymerase chain reaction and amplification of the SSR loci. The PCR procedure consisted of the following steps: (1) Initial denaturation at 94 °C for five minutes. (2) A final denaturation at 94 °C for one minute, succeeded by annealing at 65 °C for one minute and an extension phase at 72 °C for two minutes. The second step was iterated 36 times, with the annealing temperature gradually decreasing by 1 °C during each cycle until it reached a final annealing temperature of 55 °C. (3) A final extension step was carried out at 72 °C for five minutes.

### 4.4. Electrophoresis and DNA Fragment Detection

After completing the PCR, DNA electrophoresis was carried out utilizing a mini vertical electrophoresis system (MGV-202-33, CBS Scientific Company, San Diego, CA, USA). A total of 3 µL of the PCR product was mixed with 3 µL of formamide loading dye, comprising 98% formamide, 5 mM NaOH, 0.02% xylene C, and 0.02% bromophenol blue. Subsequently, 2 µL of the prepared sample was loaded onto a 6% acrylamide-bisacrylamide gel (19:1) in 0.5X tris-borate-EDTA buffer, and electrophoresis was conducted at 250 V for 40 to 60 min. Finally, the acrylamide-bisacrylamide gel was visualized through staining with ethidium bromide (EtBr) [71].

### 4.5. Statistical Analyses

The amplified DNA fragments associated with each SSR marker were recorded as either present (1) or absent (0). To collect data regarding MAF, number of alleles, GD, and PIC, we used Power Marker version 3.25 [72]. Genetic similarity was computed for every pair of accessions utilizing the Dice similarity index [73]. The similarity matrix was subsequently employed to create a dendrogram using UPGMA through the utilization of SAHN-Clustering in NTSYS-pc V2.1 [74]. Population structure analysis was performed for the 41 maize-inbred lines using STRUCTURE 2.3 software [75]. To evaluate the population structure, the membership coefficient of each cluster (K) at each subpopulation was obtained with five independent runs, with K values ranging from one to ten using an admixture model. Each run consisted of 100,000 cycles for both the burn-in and run length. The delta K (ΔK) value, which quantifies the rate of change in the log probability of data across different K values [32], was computed using STRUCTURE HARVESTER (http://taylor0.biology.ucla.edu/structharvest/, accessed on 28 July 2023). This calculation was based on the results obtained from STRUCTURE. Association mapping was conducted using TASSEL 3.0 software [76] to validate SMTAs. This analysis included two models: a general linear model (Q GLM) incorporating a population structure Q matrix and a mixed linear model (Q  + K MLM) that considered both population structure (Q) and the familial kinship (K) matrix at *p* < 0.05 [75,76]. Further, basic statistical analysis was accomplished using Microsoft Excel 365 (Version 2303), and correlation analysis for the seven physiochemical traits was performed using IBM SPSS Statistics version 21.

## 5. Conclusions

The purpose of this study was to analyze genetic diversity, population structure, and SMTAs between SSR markers and physiochemical traits in 41 maize-inbred lines to examine the genetics of these traits. The marker–trait association was performed using 200 SSR markers and seven physiochemical traits, viz., caffeic acid content, chlorogenic acid content, gallic acid content, ferulic acid content, DPPH free radical scavenging activity, LRMC, and TPC. A total of 1023 alleles were identified among 41 maize-inbred lines using the 200 SSR loci, with a mean of 5.1 alleles per locus. The average MAF, GD, and PIC were 0.498, 0.627, and 0.579, respectively. The population structure analysis based on the 200 SSR loci divided the maize germplasm into two primary groups with an admixed group. The inbred lines with membership probabilities < 0.80 were designated to an admixed group. UPGMA analysis classified the 41 maize-inbred lines into two main groups with a genetic similarity of 33%. Finally, this study identified associations between 85 SSR markers and 31 SSR markers with 7 physiochemical traits in Q GLM and Q + K MLM analyses, respectively. These SSR markers were highly correlated with at least one or more of the seven physiochemical traits. This study provides insights into the genetics of 41 maize-inbred lines and their seven physiochemical traits. In the future, the results of this study will be of assistance to breeders in marker-assisted selection for maize breeding programs.

## Figures and Tables

**Figure 1 plants-12-04092-f001:**
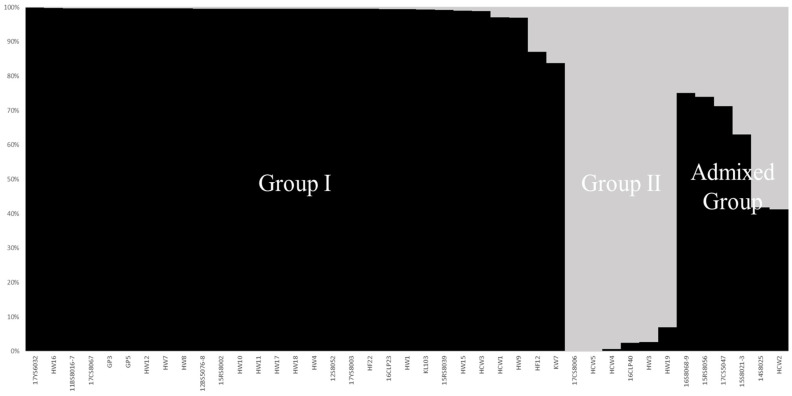
Population structure pattern in 41 maize-inbred lines based on 200 SSR markers.

**Figure 2 plants-12-04092-f002:**
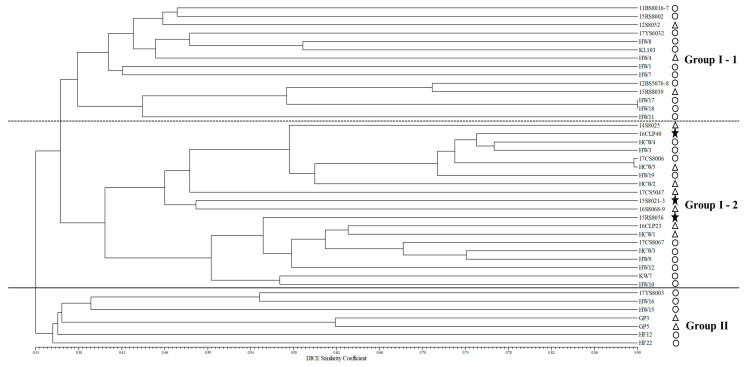
UPGMA dendrogram of 41 maize-inbred lines based on 200 SSR markers. Based on the results of seven physiochemical traits, the 41 maize-inbred lines were categorized as (○) drought-sensitive, (△) moderately drought-tolerant, and (★) drought-tolerant lines. The black solid line divides the maize population into two primary groups: Group I and Group II. However, the dotted line divides Group I into two subgroups, namely Group I-1 and Group I-2.

**Table 1 plants-12-04092-t001:** Correlation analysis for seven physiochemical traits of 41 maize-inbred lines.

	Chlorogenic Acid	DPPH	Ferulic Acid	Galic Acid	LRMC (%)	TPC
Caffeic acid average	0.103	0.063	0.296	0.150	0.181	0.548 **
Chlorogenic acid		−0.139	0.153	−0.082	0.239	0.330 *
DPPH			0.084	−0.096	−0.169	0.266
Ferulic acid				0.486 **	0.228	0.443 **
Galic acid					0.121	0.022
LRMC (%)						0.198

** Significance at *p* < 0.01. * Significance at *p* < 0.05.

**Table 2 plants-12-04092-t002:** Genetic diversity index of each chromosome for 200 SSR markers in 41 maize-inbred lines.

Chr.		No. of Alleles	MAF	GD	PIC
1 (*n* = 20)	Mean	4.6	0.570	0.570	0.521
	Min	2	0.317	0.296	0.274
	Max	8	0.829	0.779	0.747
2 (*n* = 17)	Mean	5.6	0.428	0.695	0.647
	Min	2	0.244	0.464	0.356
	Max	8	0.659	0.825	0.801
3 (*n* = 19)	Mean	5.5	0.479	0.646	0.602
	Min	3	0.220	0.396	0.354
	Max	8	0.756	0.847	0.829
4 (*n* = 18)	Mean	5.1	0.500	0.628	0.579
	Min	3	0.317	0.259	0.242
	Max	9	0.854	0.807	0.785
5 (*n* = 21)	Mean	5.5	0.515	0.623	0.579
	Min	3	0.268	0.261	0.245
	Max	10	0.854	0.823	0.802
6 (*n* = 18)	Mean	5.0	0.557	0.579	0.539
	Min	3	0.220	0.219	0.203
	Max	10	0.878	0.849	0.832
7 (*n* = 30)	Mean	5.5	0.458	0.663	0.614
	Min	3	0.244	0.290	0.260
	Max	11	0.829	0.857	0.841
8 (*n* = 20)	Mean	4.2	0.539	0.577	0.514
	Min	2	0.268	0.297	0.277
	Max	7	0.829	0.769	0.729
9 (*n* = 19)	Mean	4.9	0.472	0.633	0.590
	Min	3	0.268	0.094	0.092
	Max	7	0.951	0.789	0.757
10 (*n* = 18)	Mean	4.9	0.474	0.642	0.596
	Min	3	0.268	0.322	0.284
	Max	9	0.805	0.826	0.803
**Total**	**Mean**	5.1	0.498	0.627	0.579
**(*n* = 200)**	**Min**	2	0.220	0.094	0.092
	**Max**	11	0.951	0.857	0.841

Min: Minimum Value; Max: Maximum Value; MAF: Major Allele Frequency; GD: Gene Diversity; PIC: Polymorphism Information Content.

**Table 3 plants-12-04092-t003:** Information on SMTA markers using Q GLM for seven physiochemical traits of 41 maize-inbred lines.

Trait	Marker	Chr.	GLM	R^2^ (%)	Trait	Marker	Chr.	GLM	R2 (%)
Caffeic acid	bnlg1036	2	*	32.5		bnlg619	9	*	20.1
	umc1014	6	*	23.7		phi101049	2	*	26.6
	umc1030	3	*	31.8		umc1029	7	*	34.9
	umc1363	1	*	10.2		umc1130	8	*	12.5
	umc1863	10	*	32.1		umc1139	8	**	19.5
	umc2037	8	*	21.2		umc1177	1	*	27.4
	umc2159	5	*	28.5		umc1202	8	*	12.8
	umc2160	7	*	38.1		umc1303	4	*	19.1
	umc2185	1	*	29.1		umc1357	9	*	30.0
	umc2228	1	*	22.3		umc1401	7	**	30.3
	umc2356	8	*	24.7		umc1525	2	*	15.4
Chlorogenic acid	bnlg1209	9	*	20.9		umc1542	2	*	30.5
	bnlg1350	3	**	49.1		umc1576	10	*	18.4
	bnlg1484	1	**	35.3		umc1666	7	*	43.8
	bnlg1662	2	**	39.6		umc1709	1	*	21.7
	phi101049	2	*	26.7		umc1805	6	**	40.6
	umc1034	8	*	33.1		umc1863	10	*	28.8
	umc1065	2	*	33.0		umc1906	1	*	22.2
	umc1175	4	*	11.4		umc1949	3	*	31.7
	umc1246	10	*	12.9		umc2076	3	*	24.9
	umc1785	10	*	19.4		umc2101	3	*	27.6
	umc2026	5	*	29.9		umc2185	1	*	30.7
	umc2122	10	**	44.9		umc2224	1	**	36.6
	umc2243	1	*	22.9		umc2246	2	*	16.7
	umc2396	1	*	34.2		umc2336	9	*	31.5
DPPH	bnlg1079	10	**	41.7		umc2396	1	*	39.9
	bnlg1526	10	**	38.4	LRMC	bnlg1350	3	*	35.1
	bnlg619	9	*	21.5		bnlg339	7	**	51.5
	umc1063	6	**	50.5		phi101049	2	*	29.5
	umc1134	7	*	27.9		umc1051	4	*	27.4
	umc1380	10	*	22.7		umc1134	7	**	29.7
	umc1462	6	*	26.9		umc1239	10	*	21.1
	umc1576	10	*	16.1		umc1576	10	**	22.9
	umc1814	3	*	14.3		umc1607	8	*	35.2
	umc1887	6	*	19.6		umc1608	3	**	28.4
	umc2149	1	*	23.7		umc1627	8	**	30.5
	umc2159	5	*	21.8		umc1667	4	*	24.2
	umc2160	7	**	53.2		umc1671	7	**	36.4
	umc2217	1	*	26.5		umc1913	8	*	13.0
	umc2301	5	*	25.7		umc1934	2	*	32.7
	umc2332	7	*	15.2		umc1959	8	*	12.2
	umc2338	9	*	15.9		umc2343	9	*	23.9
Ferulic acid	bnlg1246	6	*	12.1	TPC	bnlg1350	3	*	35.1
	bnlg1759	7	*	29.0		bnlg1759	7	*	28.1
	umc1030	3	*	33.0		bnlg1887	2	**	26.2
	umc1366	9	*	16.1		phi193225	3	**	35.7
	umc1805	6	**	41.2		umc1034	8	**	43.0
	umc1906	1	**	26.3		umc1330	10	*	11.6
	umc1949	3	**	37.5		umc1388	6	**	26.5
	umc2026	5	*	33.8		umc1519	9	**	48.9
	umc2356	8	**	32.8		umc1657	9	*	28.0
Gallic acid	bnlg1450	10	*	31.9		umc1906	1	*	19.8
	bnlg1523	3	**	47.5		umc2119	3	*	22.5
	bnlg1867	6	*	34.6		umc2159	5	*	27.0

** Significance at *p* < 0.01. * Significance at *p* < 0.05.

**Table 4 plants-12-04092-t004:** Information on SMTA markers using Q + K MLM for seven physiochemical traits of 41 maize-inbred lines.

Trait	SSR Marker	Chr.	Bin	MLM		Candidate Genes/QTLs
				*p*-Value	R^2^ (%)	
Caffeic acid	umc1363	1	1.01	*	10.4	GRMZM2G095968 (*expansin-like6*) [MaizeGDB]
Chlorogenic acid	bnlg1350	3	3.08	*	46.5	-
	bnlg1484	1	1.03	*	31.4	GRMZM2G019567 (*serine/threonine-protein kinase WAG1*) [MaizeGDB]
DPPH	bnlg1079	10	10.03	*	37.4	Associated with QTLs of vanillin (Va) and syringaldehyde (Sg) traits in maize by Barrière et al. (2008) [33]
	bnlg1526	10	10.04	**	23.4	QTL for whole phosphorus utilization of plant under phosphorus deficit condition (*WPUE10a-DP*) in maize by Chen et al. (2009) [34]
	umc1063	6	6.07	*	45.6	GRMZM2G096475 (*lea protein group3*) [MaizeGDB]
	umc2160	7	7.01	*	48.1	GRMZM2G161097 (*rhicadhesin receptor*) [MaizeGDB]
	umc2301	5	5.04	*	26.9	Part of QTL associated with alanine aminotransferase (AlaAT) enzyme in maize by Zhang et al. (2010) [35]
Ferulic acid	bnlg1246	6	6.01	*	10.9	Part of the QTL cluster linked with relative above-ground biomass dry weight (RBW5-KX, RBW5-SU), relative leaf age (RLA5-KX), and tolerance index to low phosphorus (TPS5a-KX) traits in maize by Chen et al. (2008) [36]
	umc1906	1	1.05	*	22.6	GRMZM2G041050 (*nucleobase:cation symporter1*) [MaizeGDB]
	umc1949	3	3.06	*	40.4	GRMZM2G084863 (*glutaredoxin family*) [MaizeGDB]
	umc2026	5	5.05	*	32.5	GRMZM2G064255 (*glutathione transferase17*) [MaizeGDB]
	umc2356	8	8.05	*	29.1	GRMZM2G054341 (*heavy metal transport/detoxification superfamily protein*) [MaizeGDB]
Gallic acid	umc1139	8	8.01	*	11.1	GRMZM2G007915 (*small kernel 501*) [MaizeGDB]
	umc2246	2	2	*	22.1	GRMZM2G165535 (*phosphomannomutase1*) [MaizeGDB]
LRMC	phi101049	2	2.1	*	27.3	GRMZM5G850015 (*pyridoxin biosynthesis protein ER1*) [MaizeGDB]
	umc1134	7	7.03	*	26.3	GRMZM2G105253 (*Rop guanine nucleotide exchange factor 1*) [MaizeGDB]
	umc1239	10	10.03	*	19.9	GRMZM2G110145 (*Trihelix-transcription factor 43*) [MaizeGDB]
	umc1608	3	3.04	*	27.5	Linked with QTLs of total endosperm aborted kernels (EnA) and total aborted kernels (TA) traits by Qu et al. (2020) [37]
	umc1627	8	8.03	*	29.5	GRMZM2G156857 (*pathogenesis-related protein17*) [MaizeGDB]
	umc1667	4	4.08	*	26.1	GRMZM2G151044 (*trehalose-6-phosphate phosphatase4*) [MaizeGDB]
	umc1671	7	7.05	*	33.9	GRMZM2G041328 (*protein arginine methyltransferase3*) [MaizeGDB]
	umc1934	2	2.02	*	34.7	GRMZM2G045581 (*OSJNBa0070C17.17-like protein*) [MaizeGDB]
	umc1959	8	8.05	*	10.7	GRMZM2G090963 (*ATP-dependent DNA helicase Q-like SIM*) [MaizeGDB]
	umc2343	9	9.05	*	24.6	GRMZM2G078933 (*Ran-binding protein 1*) [MaizeGDB]
TPC	bnlg1887	2	2.06	*	19.8	Part of QTL associated with ear row numbers in maize by Yan et al. (2006) [38]
	phi193225	3	3.02	*	30.1	QTL for days to pollen shed (*dps3-17w*) in maize by Zhang et al. (2011) [39]
	umc1034	8	8.02	*	43.9	Part of QTL associated with gray leaf spot (*JAGLS8B*) disease in maize by Qiu et al. (2021) [40]
	umc1330	10	10.04	*	12.8	QTL for uronic acids (UA) in maize by Lorenzana et al. (2010) [41]
	umc1388	6	6.05	*	23.1	GRMZM2G093325 (related/near loci: *early response to dehydration 15-homolog3*) [MaizeGDB]
	umc1519	9	9.04	*	41.7	QTLs for root dry weight (*rdw9-1*), shoot dry weight (*sdw9-1*), and total dry weight (*tdw9-1*) under waterlogging stress in maize by Qiu et al. (2007) [42]
	umc1906	1	1.05	*	20.4	GRMZM2G041050 (*nucleobase:cation symporter1*) [MaizeGDB]

** Significance at *p* < 0.01. * Significance at *p* < 0.05.

## Data Availability

The data are contained within the article.

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
