# Peer review of "Association Mapping for Evaluation of Population Structure, Genetic Diversity, and Physiochemical Traits in Drought-Stressed Maize Germplasm Using SSR Markers"

_plants, 2023, doi:10.3390/plants12244092_

Round 1

Reviewer 1 Report

Comments and Suggestions for Authors

Dear Editor,

In the present study, the authors have tried to investigate genetic diversity in a set of hydrides in maize and applied an association mapping analysis. As this is clear to us, association mapping analysis needs a core collection and suitable plant material in terms of its size of population. In the present work, only 41 individuals were subjected to analysis, hence we can not accept the outputs. Moreover, the English language should be improved by a native person. Indeed, I found many grammatical errors in various parts of the text. Referring to the main concern of this work, I would like to encourage authors to change their objectives and only focus on investigating genetic diversity and comparing their genetic materials in terms of the measured traits. Hence, my recommendation is to "reject", and resubmit after setting up the restructuring. 

Comments on the Quality of English Language

Extensive editing of the English language required

Author Response

-> Thank you for your review comments. Although this study used 41 maize line lines as research materials for association mapping analysis, these maize inbred lines are materials that show resistance and susceptibility under drought conditions. Therefore, these resistant and susceptible maize inbred line populations are considered sufficient to perform population structure and association mapping analysis using SSR molecular markers. Particulary, in this study, we selected and analyzed a total of 200 SSR primer sets from all chromosomes to represent the entire maize genome. Therefore, the results of genetic diversity, population structure and SSR markers related to the seven traits of 41 maize line lines revealed in this study will provide useful information for future maize drought tolerance-related research. Additionally, our manuscript has been proofread in English by a native English speaking expert (see supporting materials below)

Reviewer 2 Report

Comments and Suggestions for Authors

Why did you put Materials and Methods after the Conclusion?

line 78 - It is enough to use MAS as abreviation;

line 122-125 - Which result show this?

line 126-133 - Not necessary to use the values presented in table in the text.

line 165 - Please put figure 1 after this paragraph.

line 183 - Please put figure 2 after this paragraph.

line 458-459 - How was this line obtained? By self pollination or dihaploid technique?

Author Response

-Why did you put Materials and Methods after the Conclusion?

Now, we shifted the Conclusion next to Material and Methods according to the journal format (lines 545-562).

-line 78 - It is enough to use MAS as abreviation;

Yes, you are right. So, we removed the full form and now used just an abbreviation as we already described the full form of it in the previous sentence (line 78). 

-lines 122-125 - Which result shows this?

The above-mentioned lines were a part of the UPGMA dendrogram and thus we shifted to the end of that heading (text color red) and mentioned that Fig. 2 shows this result (lines 184-187).

-lines 126-133 - Not necessary to use the values presented in table in the text.

We have removed the values from the text as per the comment (lines 123-129).

-line 165 - Please put figure 1 after this paragraph.

We have now added the figure 1 after the mentioned paragraph (lines 162-165).

-line 183 - Please put figure 2 after this paragraph.

We have now added the figure 2 after the mentioned paragraph (lines 187-191).

-lines 458-459 - How was this line obtained? By self-pollination or dihaploid technique?

All the maize inbred lines used in this study were developed through self-pollination. It’s now also added to the manuscript (lines 441-442).

Reviewer 3 Report

Comments and Suggestions for Authors

There are specific improvements should the authors consider regarding the methodology

11.  In the following sentence: ”To evaluate the population structure, we conducted five independent runs, with K values ranging from one to ten. Each run consisted of 100,000 cycles for both the burn-in and run length

-        Can you explain more this part of analysis?

2.2. You wrote: ”The acrylamide-bisacrylamide gel was visualized through staining with ethidium bromide (EtBr)

-        Which author(s) recommend EtBr to stain the acrylamide-bisacrylamide gel?

3. It is also unclear ”Admixed Group”. The 200 SSR loci divided the maize germplasm into two primary groups with an admixed group...

Author Response

There are specific improvements should the authors consider regarding the methodology

  1. In the following sentence: ”To evaluate the population structure, we conducted five independent runs, with K values ranging from one to ten. Each run consisted of 100,000 cycles for both the burn-in and run length”

- Can you explain more this part of analysis?

We have added further relevant information about the above-mentioned part of the analysis (lines 531-534).

2.2. You wrote: ”The acrylamide-bisacrylamide gel was visualized through staining with ethidium bromide (EtBr)

- Which author(s) recommend EtBr to stain the acrylamide-bisacrylamide gel?

We have now provided a suitable reference regarding the above-mentioned statement (Reference No. 71, line 522).

  1. It is also unclear ”Admixed Group”. The 200 SSR loci divided the maize germplasm into two primary groups with an admixed group...

A criteria for creating an admixed group has now been added (lines 555-556).

Reviewer 4 Report

Comments and Suggestions for Authors

The manuscript entitled Association mapping for evaluation of population structure, genetic diversity, and pysiochemical traits in drought-stress maize germplasm using SSR markers presents the findings after analyzing 41 maize inbreds with 200 SSR markers on 7 traits. The results are representing an important foundation for the marker assisted selection of the breeding programs.I congratulate the authors for their work and wish them good luck forward.

Author Response

Thank you for your appreciation.

Round 2

Reviewer 1 Report

Comments and Suggestions for Authors

The manuscript is acceptable in this form.